# Mathematical Simulation of Casson MHD Flow through a Permeable Moving Wedge with Nonlinear Chemical Reaction and Nonlinear Thermal Radiation

**DOI:** 10.3390/ma15030747

**Published:** 2022-01-19

**Authors:** Zeeshan Khan, Haroon Ur Rasheed, Ilyas Khan, Hanaa Abu-Zinadah, Maha A. Aldahlan

**Affiliations:** 1Department of Mathematics and Statistics, Bacha Khan University, Charsadda 24420, Pakistan; 2Department of Computer Science, Sarhad University of Science and Information Technology, Peshawar 25000, Pakistan; haroon.ccsit@suit.edu.pk; 3Department of Mathematics, College of Science, Al-Zulfi, Majmaah University, Majmaah 11952, Saudi Arabia; 4Department of Statistics, College of Science, University of Jeddah, Jeddah 21959, Saudi Arabia; hhabuznadah@uj.edu.sa (H.A.-Z.); maal-dahlan@uj.edu.sa (M.A.A.)

**Keywords:** mathematical simulation, RK4 method, Casson fluid, MHD porous medium, joule heating, moving wedge, unsteady flow

## Abstract

The influence of the chemical interaction and dynamic micropolar convective heat transfer flow of Casson fluid caused by a moving wedge immersed in a porous material was explored. The Joule heating owing to magnetized porous matrix heating was also deliberated. The mathematical formulation for mass conservation, momentum, energy, and concentration profiles was expressed in the form of partial differential equations. The dimensionless set of ordinary equations was reduced from modeled equations via a transformation framework and then solved by the RK4 built-in function in MATLAB SOFTWARE by taking a step size of Δη=0.01. The existing work was compared with the published work. The iteration procedure was stopped until all of the nodes in the η-direction met the convergence condition 10^−5^. The physical appearance of material parameters on the flow field, temperature, concentration, drag force, and Nusselt number was discussed through plots. The numerical results were obtained for limiting circumstances. The unsteadiness factor thinned the velocity boundary layer but decreased the thermal and concentration boundary layers. By increasing the Eckert number, the nondimensional temperature profile was enhanced. The novelty of the present study is that no one has numerically investigated the magnetized Casson fluid over a moving wedge in the presence of a chemical reaction and thermal radiation.

## 1. Introduction

The analysis and various features of non-Newtonian fluid have attracted an excessive amount of consideration due to their rigorous forms of solicitation in numerous manufacturing products and industrial goods, mostly in the abstraction of fossil fuels from crude goods, the manufacture of treacle drags, and plastic products. In the viscous and incompressible fluids class, Casson fluid has numerous choices. In this class of viscous fluid, the Casson fluid is one of which performs folding compacted at the same time, and it has the shear stress inside the critical equation for such a fluid. Non-Newtonian transmission phenomena are occurring in numerous branches of mechanics and the production of chemicals, and jointly in food processors. The different properties of the non-Newtonian liquid are exposed by some common ingredients, including ink, milk, tomato ketchup, and sugar solution. The analysis of Casson [1] defined the rheological fluid model for shade oil suspension and lithography astringent. It exhibited elastic properties and displayed the features of shear slicing, generating stress, and showed a high shear viscosity. Rao et al. [2] discussed and described innumerable features of the non-Newtonian Casson class fluid. This fluid is condensed for a non-viscous liquid in an extremely high pressure of the wall trim. The Casson liquid is illustrative and rationally evaluates the rheological effectiveness of additional liquids, along with sauces, froths, cosmetics, and biological suspensions. However, it is expected that the completion of numerous reproductions will shed light on the significance of different non-Newtonian fluids. The significant and most vital non-Newtonian fluid that has a production rate is the Casson fluid. Mitsoulis [3] elucidated stress deformation features in non-Newtonian viscoelastic fluid models and different governing flow laws in brief in 2007. However, a number of substantial flow problems that deal with compressible viscoelastic fluid flows, such as the flow around a sphere, flow from dies, and entry and exit flow, have been investigated. Various significant features of Casson liquid, such as temperature diffusion through a nonlinearly elongated stretchable sheet, have been addressed in detail in research work by Mukhopadhyay [4]. Hayat et al. [5] evaluated a substantial investigation of Soret and Dufour impacts, along with the examination of radiation on the hydromagnetic movement of Casson liquid in consideration of magnetic effects. An analytical homotopic solution was adopted and analyzed for Casson liquid due to the movement of the plate and radiation effect by Mustafa et al. [6]. In addition, both Chinyoka and Makinde [7] examined and analyzed various features of the entropy generation rate on convective unsteady fluid past a microchannel incorporated by a permeable sheet and the Navierslip. A numerical optimization was carried out for a three-dimensional nanofluid flow in a micro-channel by utilizing entropy generation minimization and a heat source in Yang et al. [8]. Slimi et al. [9] estimated the effect of entropy generation rate and the investigation of heat flow in conducting fluid within a microchannel enclosure by a permeable surface. Das and Jana [10] investigated the entropy effect in a hydromagnetic second-grade liquid stream with a permeable channel. The effects of the magnetohydrodynamic and forced convection of a CuO nanofluid flow, and the generation of the entropy, slip velocity, and temperature field were scrutinized by Abbaszadeh et al. [11]. Uddin et al. [12] explored a substantial investigation of the radiated-convective heating of nanofluid past an elongated stretching/shrinking sheet, slip effects, and porosity. Slip effects and heat convection in the nanofluid flow over a stretching surface were performed and analyzed by Shaw et al. [13]. Lopez et al. [14] explored the inspiration of entropy generation rate in the electrically conducting movement of nanoparticles over a perpendicular microchannel entrenched in a porous matrix. The magnetic influence on flow and radiations, and the analysis of convection boundary conditions with slip flow were discussed. The colloidal analysis of Casson liquid movement regarding radiation influences, the consistent/heterogeneous reactions, and the influence of a magnetic field over nonlinearly stretching sheets were explored by Khan et al. [15]. Both Vyas and Soni [16] discussed MHD effects and Casson nanofluid in a microchannel with temperature-dependent convection, entropy generation rate, and radiation analysis, and the slip effect was analyzed. The impact of focused joint density and the electro-osmotic movement of Casson fluid molten in parallel plates and their radiation properties were investigated by NG [17]. The analysis of MHD, the convective slip velocity of Casson liquid, and the entropy effect in a microchannel enclosure by porous media along with radiation effects were explored by Eegunjobi and Makinde [18]. The effect of the moment of Casson liquid in a quadrilateral chamber that covered a flow disturbed with thermal radiation was addressed in detail by Passos et al. [19].

In nuclear plants, steam turbines, solar energy generation, molten fluids, elevated plasmas, groundwater engineering, and other disciplines, the subject of hydromagnetic boundary layer movement of the position of thermal radiation arises. Applications of these kinds of fields include the melting of metal during an electric arc furnace, as well as the chilling of the first wall within a nuclear plant reactor core, where the initial plasma is segregated from the wall via a generating magnetism. Furthermore, when a maximum temperature is necessary, the effect of thermal irradiation cannot be overlooked, particularly if the existing network is housed in a thermally confined space. Furthermore, energy storage is an important worry in a number of energy-scarce places. To solve the scarcity problem, a more efficient model that can replace the current one must be devised. The liquid becomes conductive when it undergoes thermal performance, most likely as a result of the ionization induced by high temperature. The influence of heat irradiation on the convection moment of a radiated magnetohydrodynamic fluid passing a body is poorly understood. Chamkha et al. [20] considered the effects of radiation on forcible mixed-convection across a quasi-wedge under the influence of an electric field with viscous dissipation in light of its applicability. Inspired by this, Hayat and Qasim [21] used the HAM to find the solutions of magnetorheological Maxwell liquid motion due to a stretched sheet put in a porous matrix of radioactive and Joule heating. Ahmed et al. [22] premeditated the impacts of magnetism on a non-Newtonian incompressible liquid passing over a porous medium. Kandasamy et al. [23] investigated a 2D MHD boundary layer movement of Newtonian fluid filled with nanoparticles created in the influence of thermal radiation by a transparent wedge immersed in a porous media. They reached the conclusion that in the scenario of inconsistent flow, temperature is higher.

In the ferromagnetic materials, Mohyud et al. [24] inspected the heat transport in micropolar fluid. Khan et al. [25] investigated the impacts of MHD on viscous nanoparticles passing through a wedge. Khan et al. [26,27,28] addressed the heat transfer increase in an electrical conductor flow of nanofluid. In the exclusion and inclusion of nanomaterials, Ahmed et al. [29,30] scrutinized the properties of radiation with magnetic flux through a porous matrix. They used the numerical scheme to obtain numerical results. Given its extensive applications in a range of industries, the evaluation of mixed convection across crescent bodies has been intensively researched. The source of energy, petroleum products recovery, heat pipes, soil contamination, and even thermal storage are all important examples of implications of this sort of flow. Meanwhile, with the temperature field, Kumari et al. [31] explored the effect of magnetism on bioconvection over such a wedge. Gorla et al. [32] and Chamkha et al. [33] developed quasi-systems of convective heat transfer of viscoelastic fluid on the basis of a static wedge via porous media saturated with nanoparticles with and without radioactivity, respectively. Chamkha et al. [34] studied the two-dimensional convection flow caused by a wedge implanted in a porous media containing nanoparticles.

Invasive mass and operational conditions that migrate owing to movement bodies frequently cause a chemical process. A chemical process might be instantaneous or take place over a protracted era of time. Chemical reactions could be of any degree, but the smallest would be a first chemical reaction, wherein the reaction rate and the proportion of the entities are precisely proportionate. In the context of first-order reactions, Magyari and Chamkha [35] investigated nanofluids’ convection flow over a saturated porous matrix. Mukhopadhyay and Bhattacharyya [36] and Bhattacharyya et al. [37] examined the effect of first-degree chemical reactions on magnetized Maxwell and viscoelastic fluid flow on a stretched sheet. Ganapathirao et al. [38] explored the impact of reactions on transmission heat mixed convection movement via a porous segment with viscous dissipation. Khan et al. [39] used varying particle morphologies to test the moment of nanoparticles between two surfaces. The effect of chemical reactions with a mixed convection of nanofluids and Newtonian fluids was studied numerically [40,41].

Scientists, statisticians, and mathematicians, on the other hand, have recently been concerned with the analysis of boundary layer movement of non-Newtonian fluids. It is largely attributable to its extensive uses in a variety of technical and manufacturing fields. In liquid thermal performance convection and cooling operations, the understanding of these types of fluxes is critical. The normal rhythm generates a variety of properties as a result of its assortment, for example, the viscoelastic fluids model, power-law fluid model, Jeffrey fluid model, Williamson fluid model, and Eyring–Powell fluid model. The Casson model of fluid is indeed a confined rheological fluid used to explain non-Newtonian liquid movement features with a strain rate. Casson fluid has benefited a variety of industries and technical processes, including pharmaceuticals, astronautics, food production, and paper making. The Casson flow rate in crescent bodies is also useful for oil extraction, radioactive wastes’ storage, and blood vessels. Mukhopadhyay and Mandal [42] completely converted the controlling differential equation to ODEs using suitable transformation parameters and procured the numerical results of a 2D heat transmission flow through a symmetric wedge of Casson fluid based on its implementations and excellent properties. El-Dabe et al. [43] deliberated the influence of magnetism on Casson fluid caused by a flowing wedge in the vicinity of viscous dissipation and transfer of heat. Ullah et al. [44] constructed a 2D electrically conducting mixed convective flow of Casson fluid through a movable wedge soaked in a permeable material with a heat source.

The goal of this study was to learn more about the Falkner–Skan mixed convection flow through a wedge, which was inspired by the above-mentioned papers and its implementations in a variety of sciences and industries. In addition, it is clear from the literature review that the unsteady movement of Casson fluid via a rotating wedge mostly in the vicinity of a porous medium and heat generation has received very little attention. Several scholars [45,46,47,48,49] have looked at the effect of viscous dissipation in various forms of liquid dynamics in recent decades. The goal of this research was to look at how a reactive species and the heat conduction of a Casson fluid flow over a rotating wedge in the context of viscous dissipation, as well as Joule heating. Utilizing similarity transformations, all controlling nonlinear partial differential equations were converted into nonlinear ordinary differential equations, and numerical results were determined using the RK4 technique. The numerical findings for skin friction and Nusselt number were verified with published results in order to determine the correctness and certify the present results. Physically, valuable data were studied and discussed. The assumption of this investigation may lead to a strategy with more well-organized, high-grade hot rolling and glass fiber paper manufacturer machines [30].

## 2. Mathematical Modeling

In an applied magnetic field, an unsteady 2D mixed convection flow of Casson fluid over a moving wedge in the presence of porous media was examined. The wedge is moving with velocity uωx=Uωxm1−εt−12 and the free stream velocity uex=U∞xm1−εt−12, in which $U_{\omega },U_{\infty },\varepsilon \geq 0$ are constants and t is the time, where $m=\frac{\beta_{1}}{\left(2-\beta_{1}\right)},$ and $\beta_{1}=\frac{\Omega }{\pi }$ are the Hartree pressure gradient parameter and total angle of the wedge, respectively, as shown in Figure 1. In addition, a constant longitudinal magnetism $B\left(x,\;t\right)=B_{0}x^{\left(m-1\right)/2}{\left(1-\varepsilon t\right)}^{-1/2}$ is injected orthogonally toward the wedge. Due to a small Reynolds number, the induced magnetic field has been ignored. Furthermore, the temperature on that wedge wall $T_{f}\left(x,\;T\right)={\;T}_{\infty }{+T}_{0}x^{2m}{\left(1-\varepsilon t\right)}^{-2m}$ is set to the target temperature, and the concentration $C_{s}\left(x,\;T\right)={\;C}_{\infty }{+C}_{\infty }x^{2m}/{\left(1-\varepsilon t\right)}^{2m}$ with $C_{0}$ is set to the standard concentration. At streaming site $C_{\infty }{\;\mathrm{and}\;T}_{\infty },$ are the concentration and temperature respectively.

The governing equations for continuity, momentum, energy, and mass equations are [35,36]:

Continuity equation:(1)$$\frac{\partial u}{\partial x}+\frac{\partial v}{\partial y}=0$$

Momentum equation:(2)$$\begin{array}{l} \frac{\partial u}{\partial t}+u\frac{\partial u}{\partial x}+v\frac{\partial u}{\partial y}=\frac{\partial u_{e}}{\partial t}+u_{e}\frac{\partial u_{e}}{\partial x}+v\left(1+\frac{1}{\beta }\right)\frac{\partial^{2}u}{\partial y^{2}}+\left(\frac{\sigma B^{2}\left(x,t\right)}{\rho }+\left(1+\frac{1}{\beta }\right)\frac{v\phi }{k_{1}}\right)\left(u_{e}-u\right)\\ \pm \left[g\beta_{T}\left(T-T_{\infty }\right)+g\beta_{C}\left(C-C_{\infty }\right)\right]\sin\frac{\Omega }{2} \end{array}$$

Energy equation:(3)$$\frac{\partial T}{\partial t}+u\frac{\partial T}{\partial x}+v\frac{\partial T}{\partial y}=\alpha \frac{\partial^{2}T}{\partial y^{2}}-\frac{1}{\rho c_{\rho }}\frac{\partial q_{r}}{\partial x}+\frac{v}{c_{\rho }}\left(1+\frac{1}{\beta }\right){\left(\frac{\partial u}{\partial y}\right)}^{2}+\left(\frac{\sigma B^{2}\left(x,t\right)}{\rho c_{\rho }}+\left(1+\frac{1}{\beta }\right)\frac{v\phi }{k_{1}}\right){\left(u_{e}-u\right)}^{2}$$

Mass equation:(4)$$\frac{\partial C}{\partial t}+u\frac{\partial C}{\partial x}+v\frac{\partial C}{\partial y}=D\frac{\partial^{2}C}{\partial y^{2}}-k_{c}\left(C-C_{\infty }\right)$$

The extreme conditions are:(5)$$t<0:\;u=\upsilon {,T=T}_{\infty },C=C_{\infty }\;\mathrm{for}\;\mathrm{any}\;x,\;y$$
(6)$$\left.\begin{array}{c} t\geq 0:\;\mu =\mu_{\omega }\left(x,t\right)+N_{1}\upsilon \left(1+\frac{1}{\beta }\right)\frac{\partial u}{\partial y},k\frac{\partial T}{\partial y}=-h_{f}\left(T_{f}-T\right),\\ \;D\frac{\partial C}{\partial y}=-h_{s}\left(C_{s}-C\right)\;\mathrm{at}\;y=0 \end{array}\right\}$$
(7)$$\mu \to \mu_{e}\left(x,t\right),\;T\to T_{\infty },\;C\to C_{\infty }\;\mathrm{as}\;y\to \infty .$$

Here, N1x,t=N0x−m−121−εt12 stands for slip velocity, $h_{f}\left(x\right)=h_{0}x^{\frac{m-1}{2}}/\sqrt{\left(1-\varepsilon t\right)}$ and $h_{s}\left(x\right)=h_{1}x^{\frac{m-1}{2}}/\sqrt{\left(1-\varepsilon t\right)}$ are the convective heat and mass transfer equations, respectively, and $h_{0}$ and $h_{1}$ are constants. The terms $\mu_{\omega }\left(x,t\right);\mu_{e}\left(x,t\right);B\left(x,t\right);T_{f}\left(x,t\right),;C_{s}\left(x,t\right);$ and $N_{1}\left(x,t\right){;\;h}_{f}\left(x,t\right){\;\mathrm{and}\;h}_{s}\left(x,t\right)$ are lawful for $t>\varepsilon^{-1}.$ By the Rosseland estimate for thermal radiation, the heat flux $q_{r}$ is presented as [21,39,40]:(8)$$q_{r}=\frac{-4\sigma^{*}}{3k_{1}{}^{*}}\frac{\partial T^{4}}{\partial y}.$$
where $\sigma^{*}$ is the Stefan–Boltzmann constant and $k_{1}{}^{*}$ is the mean absorption coefficient. By applying the Taylor series $T^{4}$, $T_{\infty }$ can be displayed as follows:(9)$$T^{4}≅4T_{\infty }^{3}T-3T_{\infty }^{4}$$

Using Equations (8) and (9) in Equation (3), we obtain: (10)$$\begin{array}{c} \frac{\partial T}{\partial t}+u\frac{\partial T}{\partial x}+\upsilon \frac{\partial T}{\partial y}=\alpha \frac{\partial^{2}T}{\partial y^{2}}+\frac{16\sigma^{*}T_{\infty }^{3}}{3\rho c_{\rho }k_{1}{}^{*}}\frac{\partial^{2}T}{\partial y^{2}}+\frac{\upsilon }{c_{\rho }}\left(1+\frac{1}{\beta }\right){\left(\frac{\partial u}{\partial y}\right)}^{2}\\ +\left.\left(\frac{\sigma B^{2}\left(x,t\right)}{\rho c_{\rho }}\right.+\left(1+\frac{1}{\beta }\right)\frac{\upsilon \phi }{k_{1}}\right){\left(u_{e}-u\right)}^{2} \end{array}$$

We introduce the following similarity transformations:(11)$$\psi =\sqrt{\frac{2\upsilon xu_{e}}{\left(m+1\right)\left(1-\varepsilon t\right)}}f\left(\eta \right),\;\eta =\sqrt{\frac{\left(m+1\mu_{e}\right)}{2x\upsilon \left(1-\varepsilon t\right)}}y,\theta \left(\eta \right)=\frac{T-T_{\infty }}{T_{f}-T_{\infty }},\;\varphi \left(\eta \right)=\frac{C-C_{\infty }}{C_{s}-C_{\infty }},\mu =\frac{\partial \psi }{\partial y},\;\upsilon =\frac{-\partial \psi }{\partial x},$$

Using Equation (11), Equation (1) is satisfied identically, while Equations (2), (3), and (10) become dimensionless forms as:

Equation (1) is true automatically, while Equations (2)–(5) have the forms:(12)$$\begin{array}{l} \left(1+\frac{1}{b}\right)f\prime \prime \prime (\eta )+f(\eta )f\prime \prime (\eta )+\frac{2m}{m+1}\left(1-f\prime {\left(\eta \right)}^{2}\right)+\left(M+\left(1+\frac{1}{b}\right)K\right)\left(1-f\prime (\eta )\right)\\ +l\left(\theta +N\phi \right)\sin\frac{W}{2}-A\left[\frac{2m}{m+1}f\prime (\eta )+\frac{\eta }{m+1}f\prime \prime (\eta )+\frac{2}{m+1}\right]=0, \end{array}$$
(13)$$\begin{array}{l} \frac{1}{\mathrm{Pr}}\left(1+\frac{4}{3}R_{d}\right)\theta \prime \prime (\eta )+f(\eta )\theta \prime (\eta )-\frac{4m}{m+1}f\prime (\eta )\theta (\eta )+\left(1+\frac{1}{b}\right)Ecf\prime \prime {\left(\eta \right)}^{2}\\ +MEc{\left(1-f\prime (\eta )\right)}^{2}-A\left[\frac{4m}{m+1}\theta (\eta )+\frac{1}{m+1}\eta \theta \prime (\eta )\right]=0, \end{array}$$
(14)$$\begin{array}{l} \frac{1}{Sc}\phi \prime \prime (\eta )+f(\eta )\phi \prime (\eta )-\frac{4m}{1+m}f\prime (\eta )\phi (\eta )-R\phi -A\left[\frac{4m}{1+m}\phi (\eta )+\frac{1}{1+m}\phi \prime \prime (\eta )\right]=0, \end{array}$$
(15)$$\;f\prime (0)=\gamma +\delta \left(1+\frac{1}{b}\right)f\prime \prime (0),\;\theta \prime (0)=-Bi_{1}\left(1-\theta (0)\right),\;\phi \prime (0)=-Bi_{1}\left(1-\phi (0)\right),$$
where M=2σB02ρU∞1+m,A=cxU∞zm,K=2υϕk1U∞1+m,l=±GrzRex,Pr=υα,Grz=2gbTTf−T∞x3υ21+m,Rex=zuaυ,N=GraGrz,Grc=2gbCCf−C∞x3υ21+mRd=4σ*T∞3kk1*,Ec=uz2cpTf−T∞,Sc=υD,R=2υk2U∞1+m,γ=UωU∞,σ=N0m+1U∞υ2,Bi1=h0k2υU∞m+11/2,Bi2=h1D2υU∞m+11/2. represents the unsteadiness parameter, magnetic parameter, porosity parameter, thermal buoyancy parameter (λ≻0 corresponds to assisting flow, λ=0 represents nonthermal convection, and λ≺0 shows the opposing flow), Grashof number corresponding to wall temperature, Reynolds number, buoyancy ratio parameter, radiation parameter, Eckert number, Schmidt number, chemical reaction parameter, moving wedge parameter, slip parameter, and Biot numbers, respectively.

The physical quantitates reported by Mukhopadhyay and Mandal [35] are:(16)Rex12Cfx2m+1=1+1βf″0,   (Local Skin Friction)Rex−12Nux2m+1=−1+43Rdθ′0, (Nusselt number)Re−12Shx2m+1=−ϕ′0.       (Sherwood number)

## 3. Result and Discussion

The system of Equations (12)–(15) was solved numerically using the Runge–Kutta fourth-order technique. The accuracy of this approach is fourth order, and it is operationally feasible. Equations (13)–(16) were first transformed to a first-order system. The generated mathematical equations were therefore liberalized and represented in matrix form utilizing Newton’s method. Finally, using the block-tridiagonal evacuation technique, the linear set of equations was computed. In order to obtain numerical results, MATLAB software (7.10.0.499 (R2010a), The MathWorks, Inc., Natick, MA, USA) was used to create an algorithm. Furthermore, the scale size Δη and primary key $\eta_{\infty }$ for η→∞ must be assumed, and in the current study, Δη=0.01 and $\eta_{\infty }=8,10$ were employed. The convergence of the current work criteria was measured by the difference between recent and past iterations. The iteration evaluation was stopped when all of the locations in the η−direction, met the closure condition $10^{-5}$. The velocity field, temperature field, and concentration field along with skin friction and the Nusselt and Sherwood numbers were evaluated for numerous values of relevant parameters, such as unsteadiness parameter A, Casson fluid parameter b, wedge angle parameter m, magnetic parameters, porosity parameter K, thermal buoyancy parameter λ, buoyancy ratio parameter N, Prandtl number Pr, radiation parameter Rd, Eckert number Ec, Schmidt number Sc, chemical reaction parameter R, moving wedge parameter γ, slip parameter δ, and Biot numbers $Bi_{1},Bi_{2}.$ The numerically obtained results were validated with previously published work in order to determine the feasibility of the numerical procedure.

Table 1 shows the comparison of $\left(1+\frac{1}{\beta }\right)f^{\prime \prime }\left(0\right)$ (skin friction) determined by Gorla et al. [32] and Kumari et al. [31] with the present work, by giving numerous values to the parameters m, and they found great agreement among them. Table 2 associates $\left(1+\frac{1}{\beta }\right)f^{\prime \prime }\left(0\right)$ for various values of m, with the work reported by Ganapathirao et al. [38] and Mukhophadyay [42], which confirmed the validation of the proposed model. Similarly, the validation of the present results can be confirmed by comparing the Nusselt number with previously published work, as reported by Ganapathirao et al. [38] and Chamkha et al. [34] for different values of Pr, shown in Table 3.

Figure 2, Figure 3, Figure 4, Figure 5, Figure 6, Figure 7, Figure 8, Figure 9 and Figure 10 depict the variation in velocity distribution for different values of *A, m, M, K, N*, and δ, correspondingly. This can also be reduced to a Newtonian fluid by keeping β→∞ and Pr=1 fixed. Here, *m =* 0 (flat plate), *m =* 1/5 *(*Ω = 60°), *m* = 1 (stagnation point flow), *m* = 5 (Ω = 300°), and m→∞ (= 360°) are the five points of the wedge angle factor that are selected. In this study, a value of γ<0 denotes a wedge going the opposite way, a value of γ=0 denotes a static wedge, and a value of γ>0 denotes a wedge traveling in a same path. Figure 2a demonstrates the effect of A on the fluid flow. It is investigated that the fluid velocity enhances slightly as A rises. It should be noted that in ferromagnetic materials, i.e., when M≠0, the impact of A on flow rate is less prominent. The effect of b on the flow velocity for A = 0.3 is seen in Figure 2b. Increasing the amount of b lowers the velocity thickness layer. Physically, increasing b minimizes the yield stress $p_{y}$, and therefore, the thickness of the boundary layer reduces.

For different values of γ (γ>0 and γ<0), Figure 3a portrays the role of m on the fluid velocity. By increasing the values of γ, the flow rate rises only when the wedge angle varies from m=0 (flat plate) to m=5 (300°). Greater values of m shorten the boundary velocity layer thickness when γ > 0 as compared to γ < 0 situations, as shown in this diagram. In the *γ* < 0 case, the effect of m on movement is more evident. Figure 3b indicates the effect of M on velocity distribution. Fluid velocity increases in the region 0 < η < 2.5; however, the boundary layer thickness of the velocity decreases with M. This phenomenon is caused by the magnetic field parameter, which is the ratio of electromagnetism to viscous force and indicates the field strength. Furthermore, bigger values of M correlate with a high magnetic field, which increases the frictional force within the fluid, resulting in a thinner velocity boundary layer. Figure 4a demonstrates the consequence of K on the velocity field for various values of λ. It is worth noting that the fluid velocity increases with K. This increase in velocity is less noticeable in *γ* > 0 than in *γ* < 0. Actually, when the porosity of the material improves, the resistance it provides decreases. Figure 4b depicts the fluid velocity variation for numerous values of γ in the presence of a magnetic field. The velocity of the fluid decreases in opposite flows and increases in assisting flows. It is clear that strain decreases in conflicting flows, while the friction coefficient increases in aiding flows. It is also noticed that for γ > 0, the momentum boundary layer thickness declines.

The buoyancy component is the only factor in the conservation equations that ties the momentum and energy equations together. Literally, the buoyancy factor enhances the influence of the heat mutability on the speed profile, causing the flow to accelerate. Furthermore, an increase in γ causes the velocity to be influenced by convection. The influence of N on the velocity field, as seen in Figure 5a, could be explained physically in a similar fashion. Figure 5b shows how the velocity varies for altered values of L. For L > 0, the momentum boundary layer is greater than zero and it thickens, and when L < 0, the momentum boundary layer is zero and it thins. This means that if the wedge and flowing stream occur simultaneously (L > 0), skin friction reduces, but when the wedge travels in the opposite path (L < 0), the skin friction coefficient increases. As a result, it is stated that the situation of (L > 0) can be advantageous if a decrease in surface-particle friction is necessary.

Figure 6a shows the outcome of δ on the fluid flow. For increasing levels of δ, the nondimensional velocity significantly increases. The reason for this is that as δ increases, the infiltration of the fixed surface decreases, causing the velocity boundary thickness to decrease, resulting in a reduction in friction coefficient. Consequently, the boundary layer thickness is efficiently controlled by the slippage just at the wedge surface.

Figure 6b shows the influence of A and M on nondimensional heat curves. It is discovered that with improvements in A and M, the temperature enhances as the values of A and M increase. This improvement in profile has a greater impact on the free stream. Figure 5 shows the physical rationale for this behavior, which is that a larger magnetization generates a Lorentz force, which tends to increase the thickness of the thermal layer boundary.

Figure 7a represents the impact of b and m on temperature distribution. It is advantageous that the temperature distribution enhances with b. This is because viscous forces have less impact, resulting in thinner temperature and concentration boundary layers. Substantially, when the value of b increases, the viscosity of the flow decreases. Figure 7a strongly demonstrates that as b and m grow, the nondimensional temperature and volume fraction decrease. The point to be noted here is that as m increases, the temperature boundary layers become thinner.

The evolution of nondimensional temperature curves is shown in Figure 7b for numerous values of M and K. It is observed that the temperature increases with the nondimensional parameters M and K. The stimulus of γ and N on temperature distributions is shown in Figure 8a. It clearly displays that the temperature declines. As the buoyancy force leads to the friction force, intuition of λ implies a stronger rise in the flow field and, as a result, a thinner thermal boundary layer thickness is observed. A similar pattern of temperature for escalating N values is observed.

Figure 8b demonstrates the influence of γ and δ on the nondimensional temperature curve. It is clear that as γ and δ are increasing, both temperature and boundary layer thicknesses decrease. The nondimensional temperature decreases slightly as δ rises. Figure 9a depicts the rate of heat transmission for numerous values of Pr such as 0.71, 1.07, and 11.62, which physically denote air, electrolyte, and water at 400 °C, respectively. The proportion of momentum diffusion versus thermal diffusion is called the Prandtl number. It is clear that as Pr increases, the nondimensional temperature profile decreases. Thermal conductivity decreases with Pr, resulting in a thinner thermal boundary layer and lower temperature. At 400 °C, the temperature field for air $\left(Pr=0.71\right)$ and electrolyte $\left(Pr=1\right)$ decreases slowly, while for water $\left(Pr=1\right),$ it decreases quite quickly. As a result, Pr could be employed to upsurge the cooling rate in conductive fluids.

Figure 9b shows how the nondimensional temperature varies with various *Rd* and *Ec* values. Due to the involvement of heat conduction with thermal radiation heat transmission, an increment in *Rd* increases the temperature across the border region. Further, the thermal radiation, which elevates the temperature, increases the kinetic energy. It is also analyzed that as *Ec* increases, the nondimensional temperature rises. Physically, interior heating is produced as a result of viscous dissipation. As a response, the temperature enhances moderately.

Figure 10a shows the temperature curve fluctuation for different values of Bi1. Higher values of Bi1 increase the heat on the right side of the wedge surface. Physically, it makes sense as a larger Bi1 lowers the material’s thermal performance and improves convection to the flow on the right side. Figure 10b depicts the influence of Sc = 0.22 (hydrogen), Sc = 0.30 (helium), Sc = 0.62 (water vapor), Sc = 1 (CO_2_ at 250 °C), and Sc = 2.57 (propyl benzoic at 250 °C). It is worth remembering that the momentum diffusion coefficient for Sc=1 is lower than the thermal diffusivity. It is observed that the momentum and thermal radiation dissipate at the same ratio when Sc = 1. Momentum surpasses species diffusivity for  Sc=1. The nondimensional concentration profile slightly decreases with δ, as shown in Figure 10b. This is physically valid, as the slipping phenomenon allows for a further flow through the thermal and concentration boundary layers. As a result, the hot object heats a large volume of fluid, which improves thermo-physical properties. The concentration decreases with Sc and δ as displayed in Figure 11a. Sc evaluates the relative relevance of viscous and mass permittivity in terms of physics. As a result, denser species with poor thermal diffusion rate result in a large drop in the depth of the concentration boundary layer. At 250 °C and one surface pressure, the concentration for hydrogen $\left(Sc=0.22\right)$ becomes noticeably thicker than the concentration boundary layer for propyl benzoic acid $\left(Sc=2.57\right)$. Figure 11b shows how R (corresponding to a destructive chemical change) affects the concentration field. It is discovered that when R increases, the concentration decreases. The reason for this is that negative chemical reactions have a tendency to inhibit diffusion, resulting in a drop in the individual concentration’s chemical atomic diffusion rate. As a result, the concentration becomes thinner. Figure 12a shows the concentration profiles that increase with Bi2. As the temperature drives the concentration pattern, a larger Bi2 is related to better absorption of the concentration profile.

Figure 12b and Figure 13b show the Nusselt and Sherwood numbers for various values of the parameters b, k, Ec, Rd, and A, correspondingly, in addition to obtaining clarity into the change in skin friction. The deviation in the coefficient of skin friction for various values of b and k is shown in Figure 12b. It is observed that as the values of b and k enhance, the wall shear stress increases, while the friction coefficient decreases. Figure 13a shows the Nusselt number variation for several values of b and Ec. With accelerating b and Ec, the transfer of heat performance is observed to be increasing. Figure 13b shows how the Sherwood number changes with varying values of A and b. It is investigated that the mass transfer rate improves with b and A. For bigger values of b, the increase in mass transfer rate is significant.

## 4. Conclusions

The heat and mass transfer exploration for an unsteady Casson magnetic field flow through a moving wedge in a permeable sheet with nonlinear thermal radiation, chemical reaction, velocity field, as well as nonlinear convective boundary constraints was examined numerically. The basic nonlinear PDEs were transformed to nonlinear ODEs using the similarity variable and then elucidated numerically by the RK4 method. The current method was authenticated through assessment with the consequences of previous authors as a preventive case. The effects of physical factors, such as unsteadiness factor *A*, Casson parameter b, wedge angle parameter *m*, magnetic parameter M, porosity parameter K, thermal buoyancy parameter λ, buoyancy ratio factor *N*, Prandtl number *Pr*, radiation parameter *Rd*, Eckert number *Ec*, Schmidt number *Sc*, reaction factor R, moving wedge factor γ, slip parameter δ, and Biot numbers Bi1, Bi2 were deliberated graphically. The motivating conclusions of this work are:Increases in *A*, *b*, *M,* and *δ* decreased the moment boundary layer thickness.The influence of k on the flow field was more prominent in γ = 0 and γ < 0 cases.The impact of δ on the flow field was less noticeable in the presence of a magnetic field.The nonlinear thermal radiation along with concentration thickness boosted with A.Rises in λ and N intensely influenced the fluid flow; however, the influence of these factors on heat and concentration field was found slighter.The barrier yield stress was detected to decrease with b but improve with increases in A and M.The temperature and rate of heat transfer were perceived higher with the increase in Rd.The mass transfer rate was observed to be greater with the variation in chemical reaction.

## Figures and Tables

**Figure 1 materials-15-00747-f001:**
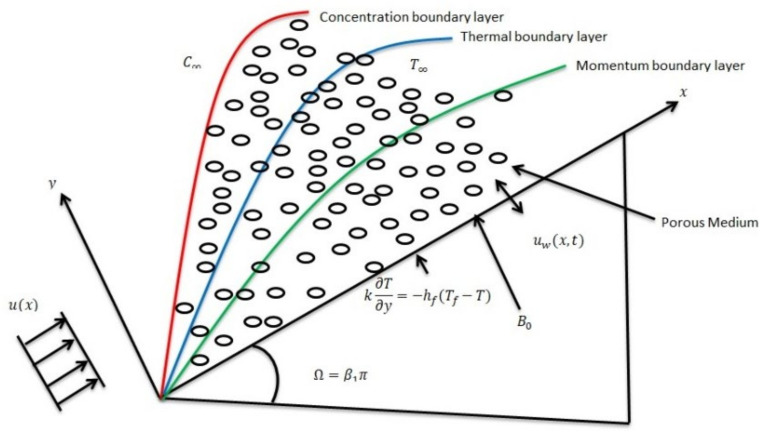
Geometry of the problem.

**Figure 2 materials-15-00747-f002:**
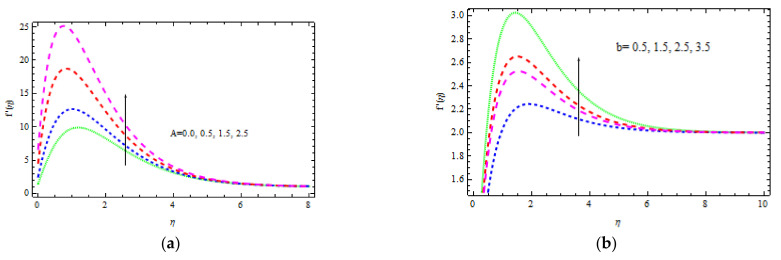
Inspiration of A (**a**) and b (**b**) on velocity.

**Figure 3 materials-15-00747-f003:**
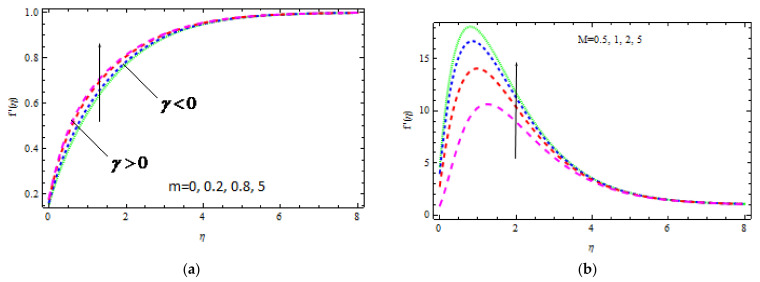
Inspiration of m (**a**) and M (**b**) on velocity.

**Figure 4 materials-15-00747-f004:**
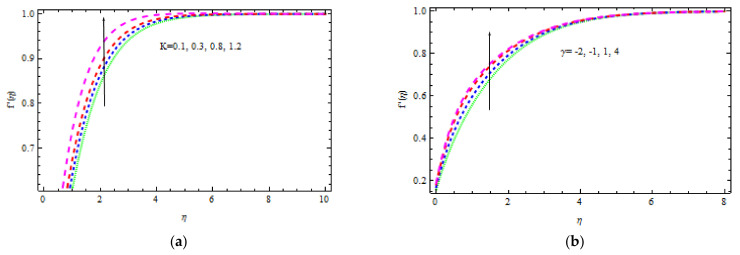
Inspiration of K (**a**) and γ (**b**) on velocity.

**Figure 5 materials-15-00747-f005:**
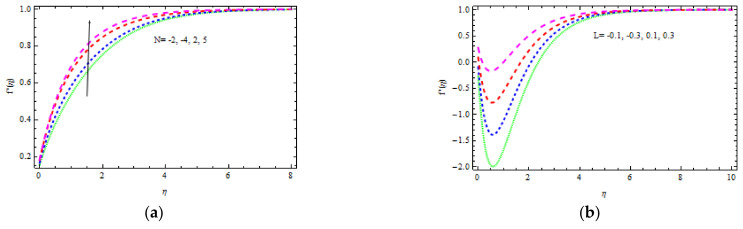
Inspiration of N (**a**) and L (**b**) on velocity.

**Figure 6 materials-15-00747-f006:**
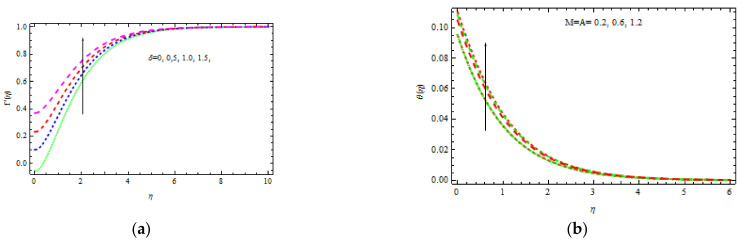
Inspiration of δ (**a**) and M and A (**b**) on velocity and temperature, respectively.

**Figure 7 materials-15-00747-f007:**
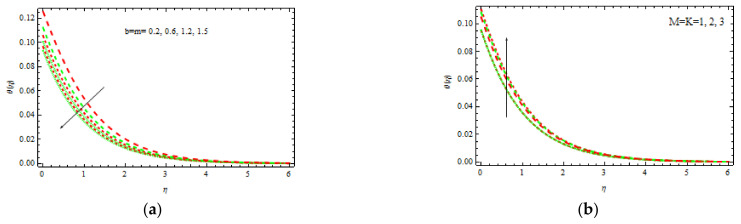
Inspiration of b, m (**a**) and M,K (**b**) on temperature.

**Figure 8 materials-15-00747-f008:**
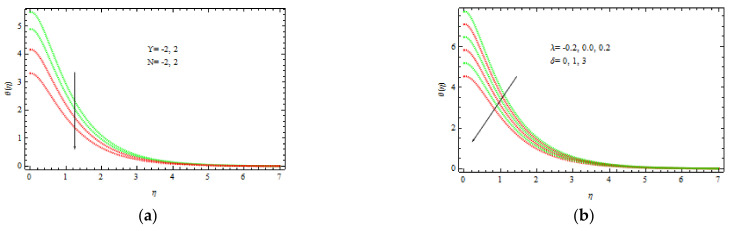
Inspiration of Υ, N (**a**) and λ, δ (**b**) on temperature.

**Figure 9 materials-15-00747-f009:**
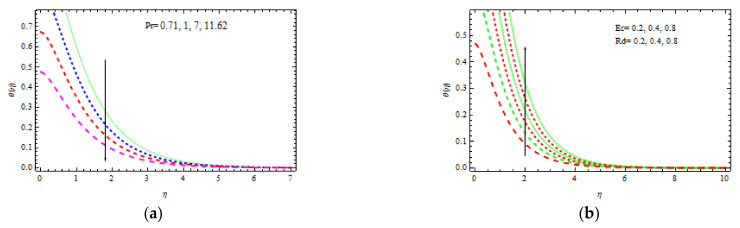
Inspiration of Pr (**a**) and Ec, Rd (**b**) on temperature.

**Figure 10 materials-15-00747-f010:**
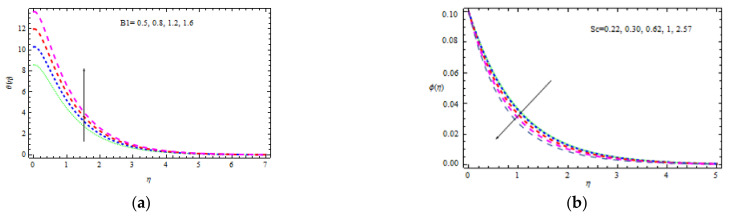
Inspiration of Bi1 (**a**) and Ec, Rd (**b**) on temperature.

**Figure 11 materials-15-00747-f011:**
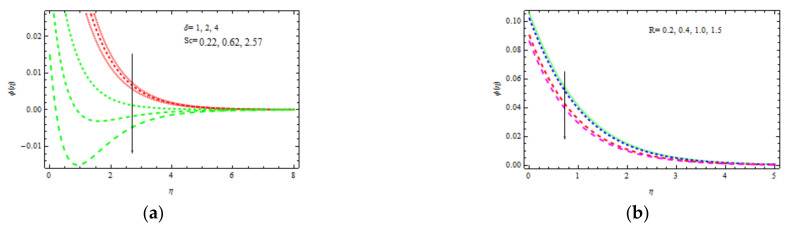
Inspiration of δ, Sc (**a**) and R (**b**) on concentration.

**Figure 12 materials-15-00747-f012:**
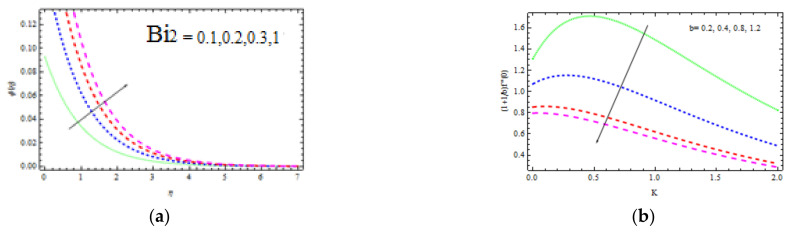
Inspiration of Bi2 (**a**) and b, K (**b**) on concentration and skin friction, respectively.

**Figure 13 materials-15-00747-f013:**
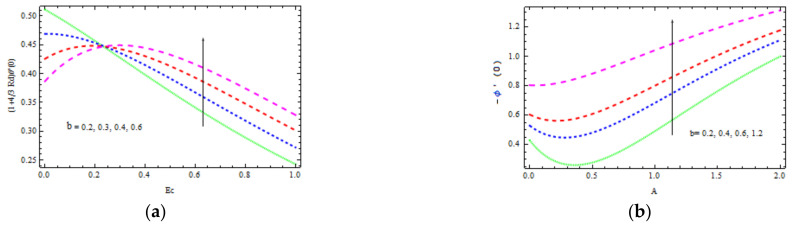
Inspiration of b, Ec (**a**) on Nusselt number and b, A (**b**) on Sherwood number.

**Table 1 materials-15-00747-t001:** Assessment of the coefficient of skin friction for numerous values of m when Pr=6.2 by taking M=K=Rd=Ec=λ=γ=δ=0,b=Bi1=Bi2→∞.

m	Gorla et al. [32]	Kumari et al. [31]	Present
0.0	0.45851	0.458506	0.45845
0.2	0.51361	0.513605	0.51356
0.34	0.55787	0.557868	0.55779
0.46	0.64387	0.643869	0.64380
1.0	0.72112	0.721117	0.72116
1.5	0.81124	0.811240	0.81116
2.0	0.91654	0.916538	0.916535

**Table 2 materials-15-00747-t002:** Assessment of the coefficient of skin friction for numerous values of m when M=K=Rd=Ec=λ=γ=δ=0 for Newtonian case.

*m*	Ganapathirao et al. [38]	Mukhophadyay [42]	Present
−0.03	0.22373	0.22365	0.22364
0.0	0.321146	0.33106	0.33103
0.33	0.747337	0.747329	0.747325
1	1.22477	1.224765	1.224762

**Table 3 materials-15-00747-t003:** Assessment of the coefficient of Nusselt number for numerous values of Pr
when M=K=Rd=Ec=λ=γ=δ=0.

*m*	Ganapathirao [38]	Chamkha [34]	Current Work
1	0.32156	0.32177	0.32147
5	0.717037	0.717051	0.717026
7	1.47075	1.47046	1.47067
15	3.2968	3.2863	3.2967

## Data Availability

The data used to support the findings of this study are included within the article.

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
