# Peer review of "Mathematical Simulation of Casson MHD Flow through a Permeable Moving Wedge with Nonlinear Chemical Reaction and Nonlinear Thermal Radiation"

_materials, 2022, doi:10.3390/ma15030747_

Round 1

Reviewer 1 Report

My general opinion about this paper is positive. My main concerns are related to section 2 that should be rewritten. Due to the lack of clarity, the provided set of formulas is hardly interpretable.

Specific comments:

- The formatting of all formulas should be corrected. Currently, their dimensions do not fit the text. I recommend using a latex template.

- Do not provide an experimental setup in the abstract. The abstract should be reorganized and rewritten. Focus it on the contribution of your paper. What problem do you address? Where is the novelty of the solution you propose?

- The literature review should be more critical. Motivate your research better.

- Section 2, each symbol used in the formulas should be defined and explained.

- Section 2 should be rewritten. Explain each formula in detail.

- It is unclear whether the system of equations provided in Section 2 was invented by you or borrowed from the literature.

- The experimental section is the best part of this paper.

Author Response

Reviewer-1

My general opinion about this paper is positive. My main concerns are related to section 2 that should be rewritten. Due to the lack of clarity, the provided set of formulas is hardly interpretable.

Specific comments:

- The formatting of all formulas should be corrected. Currently, their dimensions do not fit the text. I recommend using a latex template.

Ans. All the formatting of the formulas is written using MathType having the same dimension.

- Do not provide an experimental setup in the abstract. The abstract should be reorganized and rewritten. Focus it on the contribution of your paper. What problem do you address? Where is the novelty of the solution you propose?

Ans. The abstract contain the goal, achieving, method and solution procedure. It explains the present work. Novelty of the present work is addressed in the revised manuscript. Additionally the present work is also compared with published work for validation purpose is also explained in the abstract.

- The literature review should be more critical. Motivate your research better.

Ans. The literature has been revised. Some most relevant and recently published work has been added which are highlighted in the revised manuscript.

- Section 2, each symbol used in the formulas should be defined and explained.

Ans. It has been done accordingly using nomenclature given at the end of conclusion.

- Section 2 should be rewritten. Explain each formula in detail.

Ans. The governor equations have been explained in the revised manuscript as highlighted in red text.

- It is unclear whether the system of equations provided in Section 2 was invented by you or borrowed from the literature.

Ans. As explained in the above question, all the each formula has been explained. The basic fundamental equations such as law of conservation of mass (continuity equation), law of conservation of momentum (momentum equation), Energy equation and mass equations are first simplified. After simplification we get the highly nonlinear partial differential equations. Then by using the similarity transformations this equation are converted to ordinary differential equations. For the numerical solution, the ordinary differential equations are then converted to first order differential equations. So these are formed form the basic governors equations along with the initial and boundary conditions.

- The experimental section is the best part of this paper.

Ans. This is purely mathematical investigation. The numerical solution has been obtained by Runge-Kutta fourth order method. For the validation of the present work, the present work is also compared with published work and good agreement is found.

Reviewer 2 Report

Dear Authors,

Many thanks for your submitted work. It was interesting topic but not presented well. In total, this paper is not suitable for publication in high impact journal of materials due to several reasons which are listed as below:

  • The paper title is not new and novel.
  • The Abstract is not well written and doesnt have the structure of a good abstract. Usage of long sentences without good connection is obvious! There are no quantitative values as conclusions inside the abstract.
  • Very bad English! Finding too much errors and wrong words! Full of typos! I have never heard of MATHLAB! probably they mean MATLAB.
  • Look at such typos:
  • 250 C0
  • Prandtlnnumber, radiationpparameter,
    Eckertnnumber, Schmidtnnumber, chemicalrreaction
  • All dimensional and non-dimensional numbers are written in very bad English and even without space! I doubt that you have even dedicate a second to read what you wrote! It seems they somehow copied and pasted from somewhere! 
  • There is no novelty in your work! Please bold what is your novelty! 
  • There is no connection for Mechanical Engineering and Materials in this research! It is more or less a combination of several papers which are already published! 
  • Your thermal radiation is very simple I suppose you need to make it more advance! You need to provide enough references for your all models that you have used! 
  • Your graphs needs to be vector based all of them need refinement! 
  • Most of your conclusions are obvious! We dont need to do such work to find out these facts!
  • You need a nomenclature which represents what are the terms and the equations that you used!  

Author Response

Reviewer-2

Many thanks for your submitted work. It was interesting topic but not presented well. In total, this paper is not suitable for publication in high impact journal of materials due to several reasons which are listed as below:

  • The paper title is not new and novel.

Ans. The title of the paper has been checked through turnitin and found no similairy. In addition if any suggestion you want to give will be incorporated and will change the title.

  • The Abstract is not well written and doesnt have the structure of a good abstract. Usage of long sentences without good connection is obvious! There are no quantitative values as conclusions inside the abstract.

Ans. The abstract of the paper has been revised. It contain the goal of the paper. The fluid which is investigated has been explained. The procedure of the method has been shortly explained. The physical parameters of interest have been given. The novelty of the paper has been now explained.

  • Very bad English! Finding too much errors and wrong words! Full of typos! I have never heard of MATHLAB! probably they mean MATLAB.
  • Look at such typos:
  • 250 C0
  • Prandtlnnumber, radiationpparameter,
    Eckertnnumber, Schmidtnnumber, chemicalrreaction

Ans. The English has been improved by taking helping native English expert. For the typos, spelling and grammatical mistakes Microsoft spelling checker and grammar is also applied. Matlab is software is used. Runge-Kuuta fourth order method, finite difference method, killer-box method, ND-solve method are all the built-in-function in Matlab software. After some modification we can used these method for soling boundary value problems like boundary value problem of ordinary differential equations by converting to the first order differential equations using similarity transformations.

All these typos have been corrected in the revised manuscript.

  • All dimensional and non-dimensional numbers are written in very bad English and even without space! I doubt that you have even dedicate a second to read what you wrote! It seems they somehow copied and pasted from somewhere! 

Ans. All the dimensional and non-dimensional number are rewritten. The English has been improved. These are not copied. These are written mistakes which are now corrected in the revised manuscript.

  • There is no novelty in your work! Please bold what is your novelty! 

Ans. The novelty has been added in the revised manuscript as highlighted as a red text in the revised manuscript.

  • There is no connection for Mechanical Engineering and Materials in this research! It is more or less a combination of several papers which are already published! 

Ans. In nuclear plants, steam turbines, solar energy generation, molten fluids, elevated plasmas, groundwater engineering, and other disciplines, the subject of hydromagnetic boundary layer movement fashionable the position of thermal radiation arises. Applications of these kinds of fields include the melting of metal during an electric arc furnace as well as the chilling of first wall within a nuclear plant reactor core, where the initial plasma is segregated from the wall via generating magnetism. Furthermore, when a maximum temperature is necessary, the effect of thermal irradiation cannot be overlooked, particularly if the existing network is housed in a thermally confined space. Furthermore, energy storage is an important worry in a number of energy-scarce places. To solve the scarcity problem, a more efficient model that can replace the current one must be devised. The liquid turns conductive when it undergoes thermal performance, most likely as a result of the ionization induced by high temperature. The influence of heat irradiation in the convection moment of a radiated magneto hydrodynamic fluid passing a body is poorly understood.

The assumption of this investigation may lead to strategy more well-organized, high-grade hot rolling and glass fiber paper manufacturer machines.

  • Your thermal radiation is very simple I suppose you need to make it more advance! You need to provide enough references for your all models that you have used! 

Ans. The expression for the thermal radiation used in this investigation is the Rosseland estimate for thermal radiation. This estimation already used by many researchers like
given in references [21, 29, 38, 41] so on. Here, we mentioned the references for all the models in the revised manuscript.

  • Your graphs needs to be vector based all of them need refinement! 

Ans. All the graphs are in journal required form as in TIFF format.

  • Most of your conclusions are obvious! We dont need to do such work to find out these facts!

Ans. Here, we study the mathematical simulation in the influence of chemical interaction and dynamic micropolar convective heat transfer flow of Casson fluid caused by a moving wedge immersed in a porous material is explored. You right several authors have discussed the same phenomena. But the novelty of the present study that no one has investigates the Magnetized Casson fluid with nonlinear chemical reaction numerically by using Runge-Kutta fourth order method.

You need a nomenclature which represents what are the terms and the equations that you used!  

Ans. It has been done accordingly given below the conclusion section.

Reviewer 3 Report

I read the paper: Mathematical simulation of Casson MHD flow through a permeable moving wedge with nonlinear chemical reaction and nonlinear thermal radiation.  The paper is well written but needs the following improvements:

1- Some words in the paper are written attached together, they should be corrected. word Heat should not be CapsLock in the abstract.

2- The quality of the figures should be improved. Please provide high quality figures that do not appear fade in the paper. The legend in the figure should also be written larger. Two figures put aside each other should have the same number but with a) and b) were distinguished.

3- What is the novelty of the current work with respect to the work cited in the validation section?

4- Some industrial application of the employed test case should be provided.

Author Response

Reviewer-3

I read the paper: Mathematical simulation of Casson MHD flow through a permeable moving wedge with nonlinear chemical reaction and nonlinear thermal radiation.  The paper is well written but needs the following improvements:

  • Some words in the paper are written attached together, they should be corrected. word Heat should not be CapsLock in the abstract.

Ans. It has been corrected in the revised manuscript.

  • The quality of the figures should be improved. Please provide high quality figures that do not appear fade in the paper. The legend in the figure should also be written larger. Two figures put aside each other should have the same number but with a) and b) were distinguished.

Ans. It has been corrected accordingly. All the figures in TIFF format as required for the journal format.

  • What is the novelty of the current work with respect to the work cited in the validation section?

Ans. The novelty of the present work is now highlighted in the abstract in the revised manuscript.

  • Some industrial application of the employed test case should be provided.

Ans. As Casson fluid is non-Newtonian fluid and the application of the non-Newtonian fluid is given in the introductions section. Also the some application of such model is now given at the end of the introduction section is added.

Reviewer 4 Report

1.The original contributions need to be much better presented in the last paragraph of section “INTRODUCTION”. But the novelty of the paper is not clear. Two lines to explain the purpose of a manuscript is not sufficient! All improvements, if they are, and new results must be described in this paragraph. Please clarify the novelty of this paper with respect to published papers in the literature. 
2. Please polish the grammar. The authors should double check the mathematical formulations. A professional proofreading revision is required.
3. Please interpret the obtained results in Section “Global existence and long time asymptotic behavior”. It is
important what you conclude to them.
4. All acronyms should be defined.
5. In general, equations should be regarded as parts of a sentence and treated accordingly with the appropriate grammatical convention and punctuation. Sentences introducing equations should, with the inclusion of the equation, constitute a complete sentence. More editing for writing is needed.

6. On page 2: In the literature review , referee
would like authors review the following article about heat euation, as this helps the interested reader to be more familiar with the recent advances in this subject

https://doi.org/10.1016/j.amc.2021.126063

https://doi.org/10.1016/j.cam.2021.113695

 But, please, don't feel forced to accept the suggestions.

Author Response

Reviewer-4

1.The original contributions need to be much better presented in the last paragraph of section “INTRODUCTION”. But the novelty of the paper is not clear. Two lines to explain the purpose of a manuscript is not sufficient! All improvements, if they are, and new results must be described in this paragraph. Please clarify the novelty of this paper with respect to published papers in the literature. 

Ans. It has been incorporated in the abstract and introduction section too.

  1. Please polish the grammar. The authors should double check the mathematical formulations. A professional proofreading revision is required.

Ans. The whole manuscript is revised according to the instructions.
3. Please interpret the obtained results in Section “Global existence and long time asymptotic behavior”. It is important what you conclude to them.

Ans. It has been done in the conclusion section according to the journal required.
4. All acronyms should be defined.

Ans. All the acronyms are defined in the nomenclature given at the end of the conclusion.
5. In general, equations should be regarded as parts of a sentence and treated accordingly with the appropriate grammatical convention and punctuation. Sentences introducing equations should, with the inclusion of the equation, constitute a complete sentence. More editing for writing is needed.

Ans. It has been done accordingly.

  1. On page 2: In the literature review , referee would like authors review the following article about heat euation, as this helps the interested reader to be more familiar with the recent advances in this subject

https://doi.org/10.1016/j.amc.2021.126063

https://doi.org/10.1016/j.cam.2021.113695

 But, please, don't feel forced to accept the suggestions.

Ans. The references have been updated in the literature review relevant to the work.

Round 2

Reviewer 1 Report

The formulas still need your attention, they are poorly formatted. Correct them in the final version of the paper.

Author Response

The formulas still need your attention, they are poorly formatted. Correct them in the final version of the paper.

Ans. The formulas have been rewritten in the revised manuscript. Thank you for your kind suggestion.

Reviewer 2 Report

Dear Authors,

Many thanks for the revised version! It has been improved a lot! However, I still feel that the authors needs more work on their paper. This work is not novel! It means that there are plenty work similar to yours it doesn't matter you checked it or not! Your work is similar means just some boundary conditions have been changed! Please be more specific!

  • The quality of figures are low you must provide in a vector based version like encapsulated ones to avoid damaged figures during conversion!
  • You talked about Rosseland approximation for your work and claim many others did the same! What is your optical thickness? Please provide its amount! It doesnt matter many researchers did it! 

Author Response

Many thanks for the revised version! It has been improved a lot! However, I still feel that the authors needs more work on their paper. This work is not novel! It means that there are plenty work similar to yours it doesn't matter you checked it or not! Your work is similar means just some boundary conditions have been changed! Please be more specific!

Ans. Yes, the idea has been taken from the published work. In the published [31], the authors did not discuss the effect of nonlinear thermal radiation and nonlinear chemical reaction. In addition the boundary conditions are also changed from the published work. The published work is numerically solved by Killer box method but the present work is solved by Runge-Kutta fourth order method. So, the novelty of the present is that in the published work they did not investigate the effect of nonlinear thermal radiation and nonlinear chemical reaction and the associated boundary conditions.

  • The quality of figures are low you must provide in a vector based version like encapsulated ones to avoid damaged figures during conversion!

Ans. The quality of the figures has been improved in the revised manuscript.

  • You talked about Rosseland approximation for your work and claim many others did the same! What is your optical thickness? Please provide its amount! It doesnt matter many researchers did it! 

Ans.In the present study I told in the above question, I discussed the effect of effect of nonlinear thermal radiation and nonlinear chemical reaction. For the thermal radiation the Rosseland approximation is used which is a standard used for the thermal radiation. In the published [31] they did not discuss.

Reviewer 3 Report

The paper was improved but the quality of the figures is yet low.  Please improve the quality of the figures, export them with high quality, use annotation a, b and so on for multiple part figures.

Author Response

The paper was improved but the quality of the figures is yet low.  Please improve the quality of the figures, export them with high quality, use annotation a, b and so on for multiple part figures.

Ans. It has been done accordingly

Reviewer 4 Report

Overall Recommendation:  Accept

This manuscript is modified,
but the following comment 6  has not been covred in revised version.

  1. On page 2: In the literature review , referee would like authors review the following article about heat euation, as this helps the interested reader to be more familiar with the recent advances in this subject

https://doi.org/10.1016/j.amc.2021.126063

Author Response

  1. On page 2: In the literature review , referee would like authors review the following article about heat euation, as this helps the interested reader to be more familiar with the recent advances in this subject

https://doi.org/10.1016/j.amc.2021.126063

Ans. The reference has been added in the revised manuscript.